# Platelets and Hepatocellular Cancer: Bridging the Bench to the Clinics

**DOI:** 10.3390/cancers11101568

**Published:** 2019-10-15

**Authors:** Quirino Lai, Alessandro Vitale, Tommaso M. Manzia, Francesco G. Foschi, Giovanni B. Levi Sandri, Martina Gambato, Fabio Melandro, Francesco P. Russo, Luca Miele, Luca Viganò, Patrizia Burra, Edoardo G. Giannini

**Affiliations:** 1Department of General Surgery and Organ Transplantation, Umberto I Hospital, Sapienza University, 00161 Rome, Italy; lai.quirino@libero.it; 2Department of Surgery, Oncology, and Gastroenterology, University of Padua, 35122 Padua, Italy; alessandro.vitale.10@gmail.com (A.V.); martina.gambato@gmail.com (M.G.); francescopaolo.russo@unipd.it (F.P.R.); 3Department of Transplant Surgery, Polyclinic Tor Vergata Foundation, Tor Vergata University, 00133 Rome, Italy; tomanzia@libero.it; 4Department of Internal Medicine, Ospedale per gli Infermi di Faenza, 48018 Faenza, Italy; francesco.foschi@auslromagna.it; 5Department of Surgery, Sant’Eugenio Hospital, 00144 Rome, Italy; gblevisandri@gmail.com; 6Hepatobiliary Surgery and Liver Transplantation Unit, University of Pisa Medical School Hospital, 56126 Pisa, Italy; fabmelan@yahoo.it; 7Internal Medicine, Gastroenterology and Liver Unit, A. Gemelli Polyclinic, Sacro Cuore Catholic University, 20123 Rome, Italy; luca.miele@policlinicogemelli.it; 8Division of Hepatobiliary and General Surgery, Department of Surgery, Humanitas Clinical and Research Center, Rozzano, 20089 Milan, Italy; luca.vigano@hunimed.eu; 9Gastroenterology Unit, Department of Internal Medicine, Università di Genova, IRCCS-Ospedale Policlinico San Martino, 16132 Genoa, Italy; egiannini@unige.it; 10Associazione Italiana per lo Studio del Fegato (AISF), 00199 Roma, Italy

**Keywords:** platelet-derived growth factor, vascular endothelial growth factor, integrins, selectins, platelet-to-lymphocyte ratio

## Abstract

Growing interest is recently being focused on the role played by the platelets in favoring hepatocellular cancer (HCC) growth and dissemination. The present review reports in detail both the experimental and clinical evidence published on this topic. Several growth factors and angiogenic molecules specifically secreted by platelets are directly connected with tumor progression and neo-angiogenesis. Among them, we can list the platelet-derived growth factor, the vascular endothelial growth factor, the endothelial growth factor, and serotonin. Platelets are also involved in tumor spread, favoring endothelium permeabilization and tumor cells’ extravasation and survival in the bloodstream. From the bench to the clinics, all of these aspects were also investigated in clinical series, showing an evident correlation between platelet count and size of HCC, tumor biological behavior, metastatic spread, and overall survival rates. Moreover, a better understanding of the mechanisms involved in the platelet–tumor axis represents a paramount aspect for optimizing both current tumor treatment and development of new therapeutic strategies against HCC.

## 1. The Role of Platelets in the In Vitro/In Vivo HCC Oncogenic Process

Platelets are anucleate, disc-shaped cells present in the blood [1]. Since their discovery in 1882, our understanding of the critical role of platelets in hemostasis and thrombosis has increased remarkably [2,3].

However, beyond the role in regulating hemostasis and coagulation, multiple pieces of evidence indicate that platelets serve much more comprehensive functions in various diseases. As an example, recent discoveries revealed that platelets are also actively involved in several physiological and pathological processes, like innate and adaptive immune responses, atherosclerosis, lymphatic vessel development, angiogenesis, and carcinogenesis [4,5,6,7].

The specific correlation between platelets and cancer is a well-known condition: The first discovery of a thrombus accompanying cancer was observed over 130 years ago, and thrombocytosis —as a paraneoplastic syndrome—may occur during the natural course of neoplastic progression, frequently accompanying cancer growth and metastatic dissemination [8,9,10]. Several studies correlated a high platelet count with poor prognosis in patients with different malignancies, and, in particular, the platelet-to-lymphocyte ratio (PLR) was primarily investigated in several cancers as a useful predicting tool for patient prognosis [11,12,13].

Numerous in vitro and in vivo studies have tried to elucidate the mechanisms used by platelets in regulating carcinogenesis, tumor growth, tumor angiogenesis, tumor-related inflammation, and tumor metastasis. The number of blood-circulating platelets is 10-to-100-fold higher than that of leukocytes, therefore providing a remarkable membrane system [14]. This bioactive surface is the primary source and adaptor of numerous cross-talks between platelets and other cells. The interplay between tumor cells and circulating platelets (the so-called tumor–platelet axis) creates a reciprocal mechanism in which the tumor activates platelets, while platelets enhance tumor development and metastasis [15].

In this review, the specific connection between platelets and hepatocellular carcinoma (HCC) is reviewed, with the main intent to elucidate the different biological pathways directly involved in the HCC–platelet axis (Figure 1 and Figure 2). Moreover, from the bench to the clinics, the most recent evidence suggesting a clinical connection between HCC and platelets is reassessed, as well.

## 2. The Specific Role of Platelet-Derived Growth Factor

Currently, four members of the platelet-derived growth factor (PDGF) family have been identified: PDGF-A, PDGF-B, PDGF-C, and PDGF-D. PDGF plays an important role during embryonic development, while its role tends to decline during adulthood. However, an abnormal dysregulation of PDGF is observed in the chronic injured liver of adult patients, thus promoting liver fibrosis but also leading to sustained mito-oncogenic signaling and, therefore, favoring hepatocarcinogenesis [16].

The interaction between PDGF and its receptor PDGFR consents to the activation of the Ras/Raf/MEK/ERK pathway, one of the most relevant cellular signaling sequences in the development and maintenance of hepatocellular cancer. After the activation of Ras, this protein, in turn, activates serine–threonine kinases of the Raf-family, and activated Raf phosphorylates MEK 1/2 kinases, which eventually activate the extracellular regulated kinases ERK 1/2. Once activated, ERK 1/2 translocates to the nucleus, where it acts as a regulator of gene expression of various proteins, including those for cell cycle progression, apoptosis resistance, extracellular matrix remodeling, cellular motility angiogenesis, and drug resistance [17] (Figure 1).

Several in vitro and in vivo studies investigated the connection between PDGF and HCC (Table 1) [18,19,20,21,22,23,24,25,26,27,28,29,30,31,32,33,34,35,36,37,38,39,40,41]. In vitro studies showed a 7-fold increase in total PDGFR-alpha levels in HCC cells when compared to normal hepatocytes and a rise in chemosensitivity under normo- or hypoxic conditions when PDGF is repressed [18,19,20,21,22,24,25,26,28]. In human studies, PDGFRα overexpression is strongly correlated with HCC micro-vessel density, macroscopic vascular invasion, a shorter overall survival, and a higher HCC recurrence rate [35]. Similarly, a higher PDGF-B expression correlated with greater tumor size, a more advanced TNM stage, the presence of portal vein emboli and metastases, and worse overall survival [40]. Note that HCC patients treated with sorafenib and radioembolization reported a blunted angiogenic response following locoregional treatment [38]. Patients with initially increased PDGF levels and treated with sorafenib alone showed an improved overall survival, likely due to the ability of this drug to inhibit PDGFR [39].

More in detail, in patients with unresectable HCC who are treated with either radioembolization alone (*n* = 12) or radioembolization plus sorafenib (*n* = 11), PDGF increased compared to the baseline in the first group, while it decreased in the cases receiving sorafenib [38]. Furthermore, when comparing patients treated with hepatic artery chemotherapy (*n* = 104) vs. sorafenib (*n* = 39), patients treated with sorafenib showed a baseline PDGF-B > 300 pg/mL achieved longer survivals [39].

## 3. The Specific Role of Intra-Platelet Serotonin

Serotonin, also known as 5-hydroxytryptamine (5-HT), is a neurotransmitter mainly produced by enterochromaffin cells throughout the gastrointestinal tract, accounting for approximately 95% of its global production. Serotonin regulates a wide range of physiological and pathophysiological processes, acting as a ligand for a large family of 5-HT receptors. Platelets’ dense granules contain about 95% of total plasma serotonin, releasing it in response to different stimuli [42]. Serotonin is a potent angiogenic factor in driving tumor angiogenesis, a potent mitogen for hepatocytes, and it is crucial for liver regeneration. Recently, both in vitro and in vivo studies have shown that serotonin is involved in the tumor growth of various cancers, including HCC (Table 2) [43,44,45,46,47,48,49,50,51].

In animal models, rats pretreated with diethyl-nitrosamine showed a significant increase in serotonin, exhibiting a faster early HCC development [45]. In a zebrafish HCC model, HCC carcinogenesis was promoted by the action of serotonin-activated human stellate cells via transforming growth factor TGF-beta1 expression [50]. Lastly, several studies showed that 5-HT promotes HCC proliferation, migration, invasion, and angiogenesis through activating the Wnt/β-catenin signaling pathway (Figure 1) [44,51].

In humans, serum serotonin levels were significantly higher in cirrhotic patients with HCC as compared to those without HCC, and they were associated with a better diagnostic ability when compared with both serum alpha-fetoprotein and PIVKA-II levels (area under the curve = 0.94 vs. 0.82 and 0.92, respectively) [43]. Moreover, in 40 patients with HCC undergoing partial hepatectomy, intra-platelet serotonin levels predicted HCC recurrence (hazard ratio (HR) = 0.1, 95% confidence intervals [95%CI] = 0.01–0.89), with a disease-free interval significantly worse in patients with lower intra-platelet serotonin values [46]. Lastly, another piece of evidence supporting a potential role of serotonin in hepatocarcinogenesis was provided by two studies from Taiwan, showing that patients receiving selective serotonin reuptake inhibitors (SSRIs) due to psychological disorders had a reduced risk of HCC development [47,48]. More in detail, among 9070 HCC patients matched for age and sex with 9070 subjects without HCC, the adjusted hazard ratio for HCC in patients receiving SSRIs was 0.28 (95%CI = 0.12–0.64; *p* = 0.003); the risk was dose-dependent, with a progressive reduction observed in patients receiving higher doses of the drugs [48].

## 4. The Connection between Platelets and Epidermal Growth Factor

The epidermal growth factor (EGF) is synthesized in the megakaryocytes, being mostly present in the alpha granules of human platelets [42]. EGF is directly associated with the inflammatory microenvironment, also regulating HCC proliferation, migration, and metastatic potential of cancer cells (Figure 1 and Table 2) [52,53,54,55]. Serum levels of EGF and its receptor EGFR were found to be elevated in HCC, being potentially useful as a target for therapeutic strategies [52]. As a fact, EGFR was overexpressed in patients with HCC and cirrhosis, being detected in 33.3% of the examined HCC samples [55]. In three different animal models, the EGFR-inhibitor erlotinib reduced the receptor phosphorylation in hepatic stellate cells, also decreasing hepatocyte proliferation, liver injury, and HCC development [53]. In human HCC cell lines treated with sorafenib or regorafenib, the drug-mediated inhibition of cell growth, migration, and invasion were all antagonized by the presence of EGF and insulin-like growth factor-I, mainly when used in combination [54].

Apart the already cited sorafenib, regorafenib, and erlotinib, the drugs brivanib, gefitinib, lapatinib, imatinib, and apatinib were also introduced in clinical studies as anti-EGFR drugs, thus confirming the important role played by the pathway EGF-EGFR in the development of HCC.

## 5. The Connection between Platelets and Vascular Endothelial Growth Factor

Vascular endothelial growth factor (VEGF) represents a critical driving force for physiological and pathological angiogenesis. VEGF is largely sequestered in the platelets, and platelets act as a significant physiological transporter of this growth factor in the circulation (Figure 1) [42].

Serum VEGF levels reflect both plasmatic and intra-platelet concentrations, and since plasma contains a substantially lower level of VEGF, platelets are the primary source of serum VEGF. Several studies were published focusing on the direct correlation between serum VEGF and HCC (Table 2) [36,56,57,58,59,60,61,62,63,64]. The ratio between serum VEGF and platelet count or platelet VEGF values alone were investigated in some studies on HCC, showing a direct correlation with tumor diameter [59], recurrence after local treatment [63], vascular invasion [56,64], and overall survival [56]. A recent meta-analysis based on 11 studies evaluating the correlation between serum VEGF level and survival in patients with HCC confirmed these data, showing that serum VEGF levels had an unfavorable impact on both disease-free (HR = 2.27, 95%CI = 1.55–2.98) and overall survival (HR = 1.88, 95%CI = 1.46–2.30) [61]. Overall, the increase in serum VEGF was a poor prognostic indicator in HCC patients undergoing several different treatments, such as radiotherapy, trans-arterial chemoembolization (TACE), resection, and liver transplantation [57,58,60,62,64].

Serum VEGF derived from platelets was also investigated as a potential biomarker of sorafenib efficacy in HCC. A meta-analysis based on nine studies of HCC patients treated with sorafenib suggested that higher VEGF levels were associated with both poorer progression-free (HR=2.09; 95%CI = 1.43–3.05; *p* < 0.01) and overall survival (HR = 1.85; 95%CI = 1.24–2.77; *p* = 0.003) [63].

Several drugs showing an anti-VEGFR activity were investigated for the treatment of advanced HCC. Among them, sorafenib, erlotinib, sunitinib, brivanib, linifanib, rimacirumab, lenvatinib, regorafenib, refanetinib, and axitinib. Their great number confirms the paramount role of VEGF as the most potent angiogenic factor in HCC, therefore representing the most important target to block in HCC.

## 6. The Role of Platelets in HCC Cells Homing

Tumor metastasis to distant organs depends on the interactions between tumor cells and the host microenvironment within the circulation, lymphatic vessels, and target tissues. Platelets represent one of the bloodstream cell types contributing to metastasis. The platelet-related mechanisms favoring tumor dissemination were recently explored in more detail (Figure 2) [65]. Firstly, as previously shown, platelets play an essential role in consenting tumor growth and neo-angiogenetic remodeling in the primary tumor. These mechanisms are conveyed by the release of several platelet-related growth factors (i.e., PDGF, VEGF, and EGF) [42]. Secondly, platelets consent the tumor cells to survive in the hostile microenvironment within the bloodstream, to adhere to the endothelium, and to overpass the endothelial barrier (“extravasation”) (Figure 2) [66]. When tumor cells detach from the primary tumor and invade the blood circulation, three different factors can act as obstacles to their migration: (a) shearing forces generated by blood flow (ranging 6–15 dyn/cm^2^); (b) immunologic deletion, mainly driven by natural killer cells; and (c) anoikis, a particular type of apoptosis observed after the cell detachment from the extracellular matrix (Figure 2) [67]. Platelets can arrest all of these mechanisms thanks to the processes of (a) cohesion, (b) coagulation, (c) immune evasion, and (d) adhesion (Figure 2) [65].

During cohesion, circulating tumor cells can interact with activated platelets and leukocytes, forming hetero-aggregates that support attachment to the endothelium and thereby contribute to metastasis.

During coagulation, different receptors can be activated on tumor cells (e.g., PAR1 and PAR2), promoting the release of tissue factors (e.g., ADP, thromboxane A2, or high-mobility group box 1) further able to enhance pro-coagulant activity. This platelet activation by the circulating tumor cells represents one of the reasons for hypercoagulation and increased risks of thrombosis in cancer patients.

During immune evasion, multivalent plasma proteins form intercellular bridges, as well as activated platelets and fibrinogen, protect tumor cells from natural killer cell lysis during hematogenous metastasis. Although the exact mechanism used by platelets to protect tumor cells from immunosurveillance is still debated, two hypotheses exist, namely: (a) release of platelet-derived factors like Interferon-gamma or transforming growth factor-beta1, inducing a reduced antitumor activity in NK cells; and (b) platelet-derived molecular mimicry of circulating tumor cells favored by an MHC class I molecules expression on the tumor cell membranes.

All of these previously reported mechanisms are conveyed by the integration between platelets and tumor cells by a multitude of adhesion molecules, like GPIb-IX-V, GPVI, CLEC-2, and P-selectin. Metastasis studies indicate that platelets can use all of these molecules to support initial transient tumor cell interaction with the endothelium similarly to the recruitment of leukocytes during inflammation [68]. As an example, P-selectin mediates the aggregation of activated platelets and tumor cells, whereby the platelets can then defend the aggregated tumor cells by forming a physical barrier against the attack of circulating immune-competent cells [69].

Lastly, the transition from the initial, transient, tumor cell adhesiveness along the endothelium to the firm arrest in the metastasis site is called adhesion. This latter mechanism is mediated by the family of integrins [70]. Integrins are a family of transmembrane glycoprotein signaling receptors that can transmit bio-information bidirectionally across the plasma membrane. Integrin αIIbβ3 represents the more commonly involved integrin in cancer progression and metastasis. When active, integrin αIIbβ3 promotes outside-in signaling, which initiates and amplifies a range of cellular events to drive essential platelet functions, such as spreading, aggregation, clot retraction, and thrombus consolidation. All of these mechanisms are crucial in favoring the tumor dissemination process [71]. As for the extravasation mechanism, the activated platelets are able to product mediators disrupting adherens junctions of endothelial cells, create gaps in the endothelial monolayer, increase vascular permeability, and expose the matrix proteins of the basement membrane [66]. As an example, substances released by platelets, like eicosanoid metabolites, histamine, and serotonin, are able to regulate vessel permeability [67].

Different in vitro findings concerning the contribution of platelet count on HCC metastasis were also observed in clinical studies. In a study performed on 1613 newly diagnosed HCC patients, multivariate analysis revealed that high platelet count was associated with a higher risk of extra-hepatic metastasis (odds ratio (OR) = 4.84; 95%CI = 1.29–29.54; *p* = 0.01) [72]. Another study focused on 1660, 480, and 965 HCC patients enrolled from three hospitals showed that pretreatment platelet count (HR = 1.04 per 10,000/μL; 95%CI = 1.01–1.07; *p* = 0.01) was an independent risk factor associated with extrahepatic metastasis in early stage HCC. At ROC-curve analysis, pretreatment platelet count predicted metastasis better than alpha-fetoprotein (AFP) did, and a platelet count of <118,000/μL (HR = 0.49; 95%CI = 0.38–0.63; *p* < 0.001) or >212,000/μL (HR = 2.12; 95%CI = 1.67–2.70; *p* < 0.001) was able to categorize patients into low and high risk of metastasis subgroups [73].

## 7. The Specific Role of Platelets in the Diagnosis and Prognosis of HCC

Alterations in platelet count and function can be commonly observed in patients with chronic liver disease, and a decrease in platelet count is usually considered a noninvasive, indirect hallmark of the development of advanced disease and portal hypertension [74,75,76,77]. In patients with chronic liver disease, the pathogenesis of thrombocytopenia is multifactorial, being connected with both portal hypertension and a decrease in hepatic function [74]. Since HCC commonly arises in patients with advanced fibrosis and cirrhosis, thrombocytopenia is used in epidemiological studies with the aim to identify the population that may benefit most from HCC surveillance [78,79]. Moreover, platelet count was incorporated into models aimed at predicting the development of HCC in patients with an apparent similar risk of developing this tumor, such as patients with viral liver cirrhosis [80]. Lastly, platelet counts were included in decision-making models for selection of HCC treatment, particularly in patients with early tumors who are candidates to surgery, thus summarizing in a single parameter the information related to decrease in liver function and presence of portal hypertension [81,82,83].

As a fact, the results of a recent meta-analysis suggested that thrombocytopenia seems to be associated with reduced overall (HR = 1.41, 95%CI = 1.14–1.75) and recurrence-free survival (HR = 1.44, 95%CI = 1.13–1.83), independently from the treatment used [84]. Moreover, in another meta-analysis, thrombocytopenia was also identified as a predictor of overall (HR = 1.53, 95%CI = 1.29–1.81) and distant recurrence (HR = 1.49, 95%CI = 1.25–1.77) following any treatment [84]. These results seem to confirm the finding that low platelet count has a negative prognostic impact in patients with HCC, although several confounding factors may be difficult to disentangle from other relevant prognostic determinants [85,86].

However, there is also evidence stemming from clinical studies showing that the presence of thrombocytosis rather than thrombocytopenia may be associated with worse prognosis in patients with HCC, and that, in these patients, the presence of increased platelet count can be considered a marker of enhanced tumor aggressiveness and poorer survival [87,88,89,90]. More in detail, thrombocytosis is associated with greater tumor burden and more biological aggressiveness in patients with HCC [87,88,89,90]. Moreover, other studies have shown that an elevated pretreatment platelet count is associated with a higher risk of extra-hepatic spread in patients with early and very early HCC (HR = 2.12, 95%CI = 1.67–2.70), showing a diagnostic ability even superior in respect to AFP [72,73]. These clinical observations are supported by the basic science results previously reported, showing that platelet-derived milieu can stimulate growth and invasion of several HCC cell lines in vitro, as many mediators present in the platelets’ micro-environment are capable of enhancing tumor cell migration, invasion, and neo-angiogenesis in HCC [17,91,92].

A recent study underlined the negative effect of platelet count, activation, and aggregation in patients with nonalcoholic steatohepatitis (NASH), a pathological condition emerging as one of the leading causes of cirrhosis and HCC in Western countries. In detail, antiplatelet therapy was shown to prevent NASH and subsequent HCC development, reducing intrahepatic platelet accumulation and the frequency of platelet–immune cell interaction, thereby limiting hepatic immune cell trafficking [93].

To summarize, all these data seem to point toward a Janus Bifrons role of platelets in patients with HCC. In ancient Roman mythology, this god was portrayed with two heads facing opposite directions and represented both the beginning and the end, or, more broadly speaking, duality. Likewise, platelets in patients with HCC seem to have negative prognostic implications both when they are decreased and elevated, likely highlighting the fact that we still need to understand in more detail the role of platelets in patients with HCC.

## 8. Platelet-to-Lymphocytes Ratio as Prognostic Index for HCC

Several biomarkers were evaluated in order to assess the diagnosis, the prognosis, and the clinical management of patients with HCC. The identification of simple biomarkers able to help clinicians and oncologists in the complex scenario of liver cancer is crucial in order to optimize the clinical outcome of patients.

The link between inflammation and liver cancer was recently underlined as one of the most important pathways for cancer development, as well as being a potential target for therapy [94,95,96,97,98,99]. Tumor microenvironment is characterized by several pro-inflammatory cytokines, which can explain the onset of cancer by facilitating DNA damage, tumor growth, and angiogenesis. Thus, several pro-inflammatory markers were included in clinical scores for predicting patient survival or HCC recurrence [100,101]. The connection between platelet count and different types of solid tumors was described in several reports [102]. In the complex scenario of multiple cells and mediators involved in the connection between inflammation and tumor, platelets and their cytokines reportedly have a potential relevant role in the progression of HCC, potentially influencing the process of angiogenesis, as well as the presence of para-neoplastic events [15,103,104].

The potential availability in the clinical setting of a simple and reproducible marker for inflammation drove researchers to test the hypothesis of using pro-inflammatory markers for the identification of early stage HCC. The scores based on platelets were tested in several forms of cancer, and results are encouraging for their use as possible biomarker in clinical practice [105,106,107]. In this regard, the platelet-to-lymphocytes ratio (PLR) combined with C-reactive protein was demonstrated to be a reliable marker for small HCCs in patients with low serum AFP levels, with increasing PLR values associated with larger tumors [108,109]. Moreover, elevated PLR values were associated with poor prognosis in patients with HCC. The prognostic role of PLR was evaluated in a meta-analysis [110]. Considering the cohorts in which HCC cases underwent surgical, locoregional, or pharmacological therapies, high pretreatment PLR values were associated with a greater risk for posttreatment HCC recurrence, potentially representing a tool associated with the presence of microvascular invasion [111]. Moreover, high pretreatment PLR values were associated with both poor overall (HR = 1.73; 95%CI = 1.46–2.04; *p* < 0.001) and disease-free survival (HR = 1.30; 95%CI = 1.06–1.60; *p* = 0.01) in a recent meta-analysis [110]. In patients undergoing liver transplantation, preoperative elevated PLR values were also able to identify the risk of recurrence, with a 3.33-fold increased risk (95%CI: 1.78–6.25; *p* < 0.001) of HCC recurrence following liver transplantation [112].

## 9. Prognostic Role of Platelets in Patients with HCC Candidate to Liver Resection

The impact of platelet count on the prognosis of patients undergoing liver resection HCC was analyzed from different perspectives. The combination of thrombocytopenia and splenomegaly is an indirect sign of portal hypertension, and is associated with decreased survival following liver surgery for HCC [113].

However, portal hypertension is inherently associated with deterioration of liver function, and several studies demonstrated that, once patients are stratified according to liver function, portal hypertension is no more an independent prognostic index [114,115,116,117]. Even if the debate is still ongoing, a recent meta-analysis confirmed that indirect signs of portal hypertension (e.g., thrombocytopenia/splenomegaly) are not associated with a worse prognosis following liver resection for HCC [118]. Roayaie et al. showed that patients with a low platelet count had lower survival than patients with a normal platelet count, and that survival curves start diverging late (>2 years) after surgery, independently from postoperative recovery [119,120]. However, despite the interest in platelet count and biology in these patients, few studies analyzed the prognostic role of platelets at the molecular level. In this regard, Padickakudy et al. observed that a high preoperative intra-platelet serotonin level (>134 ng/mL) is associated with an increased risk of early tumor recurrence [121]. The same outcome (high early recurrence risk) was depicted in patients with depletion of intra-platelet serotonin after resection [46].

Considering these findings, however, the most compelling evidence is provided by studies that assessed platelet count per se (Table 3). Two meta-analyses demonstrated that platelet count is associated with lower overall survival (HR = 1.47, 95%CI = 1.21–1.78; and HR = 1.67, 95%CI = 1.22–2.27, respectively) and lower recurrence-free survival (HR = 1.36, 95%CI = 1.08–1.72; and HR = 1.44, 95%CI = 1.04–1.99) [83,122], although some limitations of the studies should be considered. First, heterogeneous cut-off values of platelet count were used in different papers. Second, many studies were performed in Eastern centers, and the negative prognostic impact of preoperative platelet count was not confirmed in a European series [122]. Lastly, results were not controlled for potential confounders, preventing the possibility to exactly define the prognostic weight of platelets.

On this topic, additional data are of interest, but need further confirmation. Shim et al. developed and validated a nomogram to predict overall and recurrence-free survivals after HCC resection, and platelet count was included into this model, although it was expressed as a continuous variable [123]. Furthermore, two studies reported that thrombocytopenia is associated with a lower risk of extra-hepatic metastases, and even though the prognostic relevance of extra-hepatic metastases is limited in HCC patients, as most recurrences are intra-hepatic, these results confirm the role of platelets in distant neoplastic diffusion highlighted for other tumors [72,73]. Finally, Pang et al. observed an inverted impact of platelet count on prognosis in non-cirrhotic patients, with thrombocytopenia being associated with lower recurrence risk [124].

Platelets were not always considered to be a single prognostic determinant, and ratios of biochemical tests, as well as formulas including platelet count, were tested as potentials predictors of prognosis in patients with HCC. Cut-off values for these variables are heterogeneous among studies, but results are concordant. The PLR, the AST-to-platelet ratio index (APRI) score, and the Fib-4 score were consistently associated with both overall and recurrence-free survivals after HCC resection, with high values being associated with poorer outcome [125,126,127,128,129,130,131,132]. Even the variation of APRI score and of the PLR in the perioperative period (delta postoperative/preoperative values) had a prognostic relevance [133,134]. When the various parameters (platelet count, APRI score and PLR) were compared, APRI score was the strongest predictor of survival [135]. Moreover, several additional scores and ratios contacting platelet values showed an association with prognosis (alkaline phosphatase-to-platelet ratio; γ-GT-to-platelet ratio; Forns index; Platelet–Albumin Score; Lok index; APGA score; PAPAS score), although they need further confirmation [124,136,137,138,139,140].

Lastly, some authors from Asian countries proposed to treat patients with HCC and thrombocytopenia by combining liver resection and splenectomy, and this approach was associated with favorable survival results [141,142,143]. More in detail, in one of these studies, patients with portal hypertension undergoing liver resection and splenectomy had even higher survival rates than patients with portal hypertension undergoing liver resection alone [143]. In a recent study, a high splenic volume was an independent predictor of overall and recurrence-free survivals, with patients presenting a baseline high splenic volume undergoing splenectomy reaching better survivals in respect to patients with “untreated” splenomegaly [144]. The possibility of correcting thrombocytopenia, thus abolishing its negative prognostic impact, is fascinating, but needs stronger evidence.

## 10. Prognostic Role of Platelets in Patients with HCC Candidate to Liver Transplant and in Liver-Transplant Recipients

Liver transplantation is the treatment for patients with cirrhosis and HCC according to well-reported selection criteria, offering the opportunity to treat the tumor and the underlying liver disease at the same time [145]. In the last 20 years, several selection criteria for HCC were developed on pre-transplantation features or explant pathology in order to optimize overall patient survival after transplantation [146,147,148]. Most of these criteria include tumor parameters, as well as biology surrogates, such as serum AFP and response to pre-transplantation neoadjuvant therapies [149]. Other markers of microvascular invasion—known to be associated with a high rate of HCC recurrence after transplant—were investigated, including the systemic-inflammation parameters neutrophil-to-lymphocyte ratio and PLR (Table 3) [111]. In a study performed on 181 HCC patients listed for liver transplantation, of whom 146 were undergoing liver transplantation, PLR showed an intermediate capacity to predict the post-transplant HCC recurrence, being also able to stratify patients in relation to tumor-free survival. The study concluded that the use of this easily available marker might represent an additional tool in the selection of patients with HCC undergoing liver transplantation [111]. Another study investigated the prognostic value of the PLR in 70 patients with HCC treated with TACE and waiting for a liver transplant, of whom 31% were outside Milan Criteria and 17% outside UCSF Criteria [150]. The Authors reported that the mRECIST nonresponse to TACE, exceeding UCSF criteria before TACE and a preoperative PLR > 150, were independent predictors of tumor recurrence. Similarly, an Asian group performed a retrospective analysis on 343 patients transplanted for HCC by using the PLR to stratify patients exceeding Hangzhou Criteria [151]. The group identified 120 as the most significant cut-off value when comparing recurrence-free survival of patients exceeding the Milan but fulfilling the Hangzhou Criteria: after stratification based on PLR < 120, the one-, three-, and five-year recurrence-free survivals were 84%, 73%, and 73%, respectively. On the opposite, patients with a PLR > 120 showed poor outcomes, with one-, three-, and five-year recurrence-free survivals of only 37.5%, 12.5%, and 12.5%, respectively [151]. Lastly, multivariate analysis confirmed that a PLR ≤ 120 was independently associated with recurrence-free survival in patients exceeding the Milan but fulfilling the Hangzhou Criteria, confirming a previous study from the same group, where a PLR ≤ 125 was associated with a more advanced tumor stage and more aggressive tumor behavior [152].

In regard to the discussion about the appropriate cut-off value of the PLR, a recently published meta-analysis including 899 cases showed that the cut-off value of 150 was reported to be the best in four out of five selected papers fulfilling the selection criteria of the study [112]. However, some studies did not confirm the role of this surrogate marker of HCC recurrence after transplantation. In one of these studies, 150 consecutive MC-IN transplanted patients did not show any association between PLR and post-transplant HCC recurrence or worse overall survival [153].

An interesting recently published study evaluated the association between preoperative platelet count and HCC recurrence in 359 patients undergoing living-donor liver transplantation for HCC [154]. A total number of 209 patients who had a preoperative platelet count of ≤ 75 × 10^9^/L were matched with 97 patients who had a preoperative platelet count of > 75 × 10^9^/L. Recurrence risk was significantly greater in the high-platelet group in both univariate and multivariate analyses. Preoperative platelet did not interact with the Milan Criteria, AFP level, Edmonson grade, microvascular invasion, or intrahepatic metastasis. However, incorporation of platelet into the Milan Criteria significantly improved predictive power, and PLR did not show superiority to platelet count alone into the Milan Criteria in predicting HCC recurrence after transplant.

In summary, it appears from the majority of published studies that there is a correlation between PLR and HCC aggressiveness in the setting of liver transplantation. High pre-transplant values may cause up to a 3.3-fold increased risk for post-transplant recurrence. PLR is an easy and inexpensive parameter that can be used to select patients with hepatocellular cancer waiting for liver transplantation. However, more studies aimed at better understanding biological and clinical mechanisms of the link between PLR and HCC are needed before being able to draw definitive conclusions [112].

## 11. Conclusions

The growing relevance that the platelets show to play in the mechanisms connected with tumor progression and metastatic spread was shown by the numerous articles focused on the different pathways the platelets concur in favoring. The specific role of platelets in consenting the tumor cell homing is clear. All of these mechanisms were also confirmed in the specific setting of hepatocellular cancer. From the bench to the clinics, the role of thrombocytosis and platelet-to-lymphocyte ratio was largely demonstrated in HCC patients undergoing locoregional therapies, surgical resection, and liver transplantation. Further evidence is required with the intent to optimize the current prognostic evaluation of the patients and to identify new therapeutic strategies aimed at blocking the pathways in which the platelets are involved.

## Figures and Tables

**Figure 1 cancers-11-01568-f001:**
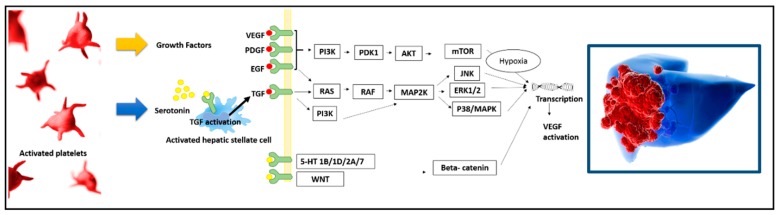
Molecular mechanisms connecting platelets and hepatocellular cancer (HCC), supporting tumor local progression.

**Figure 2 cancers-11-01568-f002:**
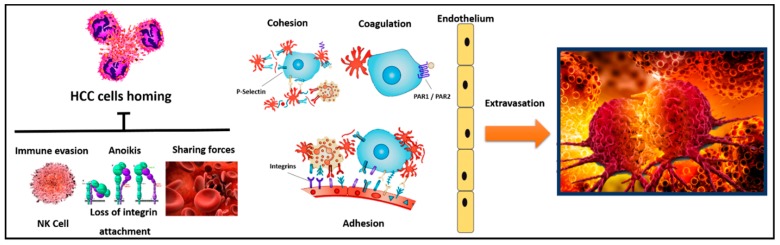
Molecular mechanisms connecting platelets and HCC, supporting tumor metastatic spread from the bloodstream.

**Table 1 cancers-11-01568-t001:** Studies focused on the role of platelet-derived growth factor in the development of HCC.

Year	Author	Results
In Vitro		
2007	Stock P [18]	63% of HCC tissues showed up to 7-fold increase in total PDGFRa levels compared with adjacent tissue controls; anti-PDGFa decreases cell proliferation in HCC cell lines.
2009	Lau CK [19]	HCC cell lines and hepatic progenitor cell lines treated with cisplatin under normo- or hypoxic conditions; blockade of the Akt/HIF-1α/PDGF-BB autocrine signaling increases the chemosensitivity of HCC cells and hepatic progenitor cells under hypoxic conditions.
2012	Okada H [20]	Peretinoin represses the expression of PDGF-A/B in primary mouse hepatoma cells, preventing the progression of fibrosis and the subsequent development of HCC.
2013	Wu Q [21]	Gemcitabine-resistant HCC cells; PDGF-D highly expressed in these cells, with down-regulation of PDGF-D leading to partial reversal of the epithelial-mesenchymal transition.
2015	Hara Y [22]	Multikinase-inhibitor TSU-68 inhibits stromal PDGF signaling activated by HCC cells and inhibits HCC growth.
2015	Wang R [23]	PDGF-D is highly expressed in gemcitabine-resistant HCC cells.
2015	Lu Y [24]	Hepatic stellate cells in hypoxic condition; PDGF-BB expression markedly increased, while PDGF-BB blocking abolished cell proliferation, migration, and vascular endothelial growth-factor-A expression.
2016	Cho Y [25]	Human HCC cell lines cocultured with activated human hepatic stellate cell line under normo- or hypoxic conditions; hypoxic stellate cell-derived PDGF-BB stimulates the proliferation of HCC cells through activation of the PI3K/Akt pathway, while the inhibition of PDGF-BB or PI3K/Akt pathways enhances apoptotic cell death.
2016	Ma Y [26]	Human HCC cell lines; insulin-like growth factor-binding protein-3 suppresses PDGF expression.
2017	Lv X [27]	Cocultured hepatic stellate cells and HCC; PDGF is an effective activator of hepatic stellate cells.
2019	Xiao Z [28]	HCC cells; XPD suppresses cell proliferation and migration via regulating miR-29a-3p-Mdm2/PDGF-B axis
In Vivo (animal models)		
2007	Campbell JS [29]	Transgenic PDGF-C mice with HCC; imatinib treatment decreases PDGFRa and stromal cell proliferation.
2011	Maass T [30]	Transgenic PDGF-B mice with HCC; PDGF-B mice present HCC larger than wild-type.
2012	Zhang JB [31]	HCC nude mouse model; PDGF-A up-regulated when IFN-α treatment re-initiated.
2014	Wright JH [32]	Mouse model; PDGF-C induces progressive fibrosis, chronic inflammation, neoangiogenesis, sinusoidal congestion, and global changes in gene expression.
In Vivo (Human)		
2007	Mas VR [33]	From the gene expression analysis of the HCV–HCC tumors compared to normal livers, an important number of genes related to angiogenesis was differentially expressed, including PDGF. PDGF was also statistically differentially expressed between HCV cirrhosis and HCV–HCC groups.
2013	Chen YW [34]	HCC cases = 21 vs. controls = 8; circulating PDGF not higher in HCC.
2014	Wei T [35]	HCC = 57 vs. adjacent nontumor tissue = 57; PDGFRα overexpression strongly correlated with HCC microvessel density (*p* < 0.05), macroscopic vascular invasion (*p* < 0.05), shorter overall survival, and higher recurrence rate (*p* < 0.05, respectively).
2014	Talaat RM [36]	HCC cases = 135 vs. healthy cases = 50; higher PDGF plasma levels observed in HCC vs. healthy controls (*p* < 0.001).
2015	Alkozai EM [37]	HBV/HCV with or without HCC = 38, healthy volunteers = 20; intraplatelet and plasma levels of PDGF comparable between patients and controls.
2016	Lewandowski RJ [38]	Unresectable HCC treated with TARE alone = 12 or TARE + sorafenib = 11; In TARE/sorafenib group, PDGF decreased, in TARE only PDGF increased respect to baseline (*p* = 0.03).
2017	Hayashi T [39]	Hepatic arterial infusion chemotherapy = 104 vs. sorafenib = 39; patients treated with sorafenib with higher serum PDGF-BB (>300 pg/mL) achieved longer survival.
2018	Chen B [40]	HCC after hepatectomy = 90; higher PDGF-B expression correlated with tumor size (*p* = 0.02), TNM stage (*p* = 0.047), and portal vein emboli and metastases (*p* = 0.04). Higher PDGF-B is associated with worse survival (*p* = 0.002).
2019	Aryal B [41]	HCC patients undergoing resection = 40; lower serum PDGF-BB independent predictor of HCC recurrence after hepatic resection (HR = 5.64, *p* < 0.01).

Abbreviations: HCC, hepatocellular cancer; PDGFR, platelet-derived growth factor receptor; PDGF, platelet-derived growth factor; Akt, protein kinase B; HIF-1α, hypoxia-inducible factor 1-alpha; PI3K, posphoinositide-3 kinases; IFN-α, interferon alpha; HBV, hepatitis B virus; HCV, hepatitis C virus; TARE, trans-arterial radioembolization; TNM, tumor node metastasis; HR, hazard ratio.

**Table 2 cancers-11-01568-t002:** Studies focused on the role of serotonin, epidermal growth factor (EGF), and vascular endothelial growth factor (VEGF) in the development of HCC.

Year	Author	Results
Serotonin		
2015	Fatima S [43]	HCC cell lines and 33 pairs of HCC and corresponding adjacent non-tumor tissues. Receptors 5-HT1D (21/33, 63.6%), 5-HT2B (12/33, 36.4%), and 5-HT7 (15/33, 45.4%) were overexpressed. Serotonin increased total b-catenin and active b-catenin, and decreased phosphorylated b-catenin protein levels.
2016	Abdel-Razik A [44]	HCV–cirrhosis + HCC (*n* = 82), HCV–cirrhosis (*n* = 80), chronic HCV (*n* = 100), healthy controls (*n* = 60). Serotonin levels higher in cirrhotic vs. chronic HCV cases (*p* < 0.001) and in HCC vs. only cirrhosis cases (*p* < 0.001). HCC diagnosis better using serotonin vs. AFP or PIVKA (AUC 0.94 vs. 0.82 and 0.92).
2016	Abdel-Hamid NM [45]	HCC rat models. Significant increase in serotonin. Only serotonin exhibited a significant increase in early histological stage HCC development.
2017	Aryal B [46]	40 HCC patients undergoing partial hepatectomy. Intra-platelet serotonin levels predicted HCC recurrence (HR = 0.1, 95%CI = 0.01–0.89). Disease-free interval significantly worse in patients with low intra-platelet serotonin (*p* = 0.029).
2017	Chan HL [47]	Taiwan’s National Health Insurance Research Database included 59,859 HCC cases vs. 285,124 matched controls. SSRIs associated with lower HCC risk, and the findings were dose-dependent (*p* < 0.001).
2017	Chang CM [48]	Taiwan’s National Health Insurance Research Database 9070 HCC vs. non-HCC subjects analyzed after matching for age and sex. HR for HCC in patients with SSRI use was 0.28 (95%CI = 0.12–0.64; *p* = 0.003). For SSRI users with a cumulative defined daily dose of 28–89, 90–364, and ≥365, HRs 0.51, 0.22, and 0.12.
2017	Liu S [49]	Human HCC cell lines. Yes-associated protein promoted by serotonin, favoring cell proliferation, invasion, and metastasis.
2017	Yang Q [50]	Zebrafish HCC model. Serotonin-activated human stellate cells promote HCC carcinogenesis and increase serotonin synthesis via transforming growth factor TGFb1 expression, hence causing a sex disparity in HCC (more tumor cases in male fishes).
2019	Zuo X [51]	96 pairs of HCC and peritumor samples from resected patients. Serotonin 1D expression level significantly up-regulated in HCC tissues and cell lines, closely correlating with unfavorable clinicopathological characteristics.
EGF		
2014	Huang P [52]	Cell bio-behaviors of HCC with low or high metastasis detected by live cell monitoring system. EGF significantly induced cell proliferation in HepG2 cells. EGF prompted cell movement in both HepG2 and HCCLM3 and regulated the production of CXCL5 and CXCL8 from HCC, which were inhibited by EGFR inhibitor, Erk inhibitor (U0126), or PI3K inhibitors.
2014	Fuchs BC [53]	Three different HCC animal models: rat model induced by diethylnitrosamine, mouse model induced by carbon tetrachloride, and a rat model induced by bile duct ligation. Erlotinib reduced EGFR phosphorylation in hepatic stellate cells, also decreasing hepatocyte proliferation and liver injury. Erlotinib also blocked the development of HCC.
2014	D’Alessandro R [54]	Human HCC cell lines with or without Sorafenib/Regorafenib. Drug-mediated inhibition of cell growth, migration, and invasion were all antagonized by platelet lysates. EGF and insulin-like growth factor-I able to antagonize Sorafenib in a proliferation assay, particularly in combination.
2015	Badawy AA [55]	40 core liver biopsies from patients with HCV, 20 liver specimens from HCC cases with HCV, and 5 normal controls. EGFR and TGF-α were overexpressed in HCC and cirrhotic cases compared to HCV cases without cirrhosis. EGFR was detected in 33.3% of the examined HCC cases.
VEGF		
2004	Kim SJ [56]	52 HCC, 26 liver cirrhosis patients and 30 healthy controls. Serum VEGF per platelet count was higher in HCC than in liver cirrhosis patients and healthy controls (*p* < 0.01). Statistically significant correlation between serum VEGF and platelet count in HCC patients. Serum VEGF per platelet count higher in patients with advanced-stage and portal-vein thrombosis (*p* < 0.01). Patients with high serum VEGF per platelet count had poor response to treatment and shorter overall survival (*p* < 0.01). Serum VEGF per platelet count independent prognostic factor for the presence of portal vein thrombosis (*p* < 0.01).
2009	Hu J [57]	162 AFP-negative HCC patients undergoing curative resection. Positive rates of VEGF and PD-ECGF in tumor tissues were 59.9% and 62.3%. At multivariate analysis, VEGF/PD-ECGF index independent prognostic factor for overall survival and relapse-free survival (*p* = 0.002 and *p* < 0.001).
2009	Corradini SG [58]	24 patients undergoing liver transplant. VEGF-A more expressed in HCC than in non-cirrhotic tissue (*p* < 0.05).
2011	Ferroni P [59]	HCC (*n* = 70), cirrhosis (*n* = 45), and control subjects (*n* = 70). Median concentrations of plasma VEGF/platelet higher in HCC or cirrhotic patients compared to controls (*p* = 0.002). VEGF/platelet-load correlated with tumor diameter (*p* < 0.05).
2012	Guo JH [60]	60 HCC patients undergoing TACE or transarterial infusion for unresectable tumor vs. 12 healthy volunteers. Median serum VEGF level in the HCC patients significantly higher than that of healthy controls (*p* = 0.021). Serum VEGF levels significantly correlated with platelet counts. Patients with serum VEGF level >285 pg/mL had worse overall survival (*p* = 0.002). By multivariate analysis, the serum VEGF level was a significant prognostic factor.
2013	Zhan P [61]	Meta-analysis of 11 studies evaluating the correlation between serum VEGF level and survival in patients with HCC. Combined hazard ratios suggested that serum VEGF level had an unfavorable impact on overall survival (HR = 1.88, 95%CI: 1.46–2.30), and disease-free survival (HR = 2.27, 95%CI: 1.55–2.98) in patients with HCC.
2014	Talaat RM [36]	135 HCC patients (57 Child-Pugh A, 24 Child-Pugh B, and 54 Child-Pugh C stage) and 50 healthy subjects. Significant increase in plasma levels of VEGF (*p* < 0.001), PDGF (*p* < 0.001), TNF-α (*p* < 0.01) in HCC patients. Maximum production of VEGF and TNF-α was present in Child-Pugh C patients.
2014	Suh YG [62]	50 HCC patients treated with radiotherapy. Patients with recurrence outside the radiation field had higher VEGF-A/platelet levels before and after radiotherapy (*p* = 0.04). On multivariate analysis, high level of VEGF/platelet before radiotherapy significant independent prognostic factor for a worse progression-free survival (*p* = 0.04).
2015	Cao G [63]	Meta-analysis based on 9 studies evaluating the relationship between VEGF level and clinical outcome in advanced HCC patients treated with sorafenib. Pooled estimates suggested that high level of VEGF was associated with poor overall survival (HR = 1.85; 95%CI: 1.24–2.77; *p* = 0.003) and poor progression-free survival (HR = 2.09; 95%CI: 1.43–3.05; *p* < 0.01).
2016	Aryal B [64]	37 HCC resected patients. Serum and intra-platelet VEGF-A significantly elevated at four weeks of resection. Preoperative intra-platelet VEGF-A higher in patients with advanced cancer and vascular invasion. Postoperative intra-platelet VEGF-A higher after major liver resection.

**Abbreviations:** HCC, hepatocellular cancer; 5-HT,5-hydroxytryptamin; HCV, hepatitis C virus; AFP, alpha-fetoprotein; PIVKA, protein induced by vitamin K antagonism; AUC, area under the curve; HR, hazard ratio; SSRI, selective serotonin re-uptake inhibitors; EGF, epidermal growth factor; VEGF, vascular endothelial growth factor; PD-ECGF, platelet-derived endothelial cell growth factor.

**Table 3 cancers-11-01568-t003:** Studies reporting the negative clinical impact in HCC patients showing high platelet count in terms of tumor-free or overall survival rates after curative treatments.

Year	Author	Survivals
Resection		
2014	Shen SL	5-year tumor-free survival: APRI < 0.62: 32% 5-year tumor-free survival: APRI ≥ 0.62: 19%
2015	Ni XC	2-year survival: PLR < 150: 90% 2-year survival: PLR ≥ 150: 77%
2016	Ji F	5-year tumor-free survival: APRI < 1.68: 38% 5-year tumor-free survival: APRI ≥ 1.68: 21%
2016	Goh BK	1-year mortality: PLR < 290: 13% 1-year mortality: PLR ≥ 290: 34%
Liver Transplantation		
2013	Lai Q	5-year tumor free survival: PLR < 150: 89% 5-year tumor free survival: PLR ≥ 150: 50%
2015	Xia W	5-year tumor free survival: PLR < 150: 92% 5-year tumor free survival: PLR ≥ 150: 81%
2016	Harimoto N	5-year tumor free survival: PLR< 150: 52% 5-year tumor free survival: PLR ≥ 150: 25%
2017	Nicolini D	5-year tumor free survival: PLR < 150: 95% 5-year tumor free survival: PLR ≥ 150: 76%

Abbreviations: APRI, AST to platelet ratio index; PLR, platelet-to-lymphocyte ratio.

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
