# Peer review of "Platelets and Hepatocellular Cancer: Bridging the Bench to the Clinics"

_cancers, 2019, doi:10.3390/cancers11101568_

Round 1
Reviewer 1 Report
The present review deals with the current knowledge of the impact of platelets on hepatocellular cancer progression and metastasis with special focus on PDGF, serotonin, EGF, and platelets as biomarkers in different clinical settings.
Major issues:
- The mechanisms by which platelets support cancer metastasis could be explained in more detail.
-figure 1: different pictures are integrate in the figure but the intention and content are hard to follow or to understand. In the introduction you refer to figure 1 but the different signaling pathways in the figure are not mentioned in the text. The figure should be modified completely and presented clearer, for instance, it is not clear where Growth Factors and Serotonin come from in the upper part of the figure.
-line 50: HCC not Hcc
-line 97: maybe one space character to much between PDGFR and levels
-line 100-103: you don´t need to indicate the p-values in a review article, that is rather uncommon.
-line 104-107: the sentence is very long and difficult to understand. Please divide into two sentences.
-line 112: delete “who”
-Table 1: maybe you should not label it “learning poin”t, rather “result” or “outcome”.
-Table 1: 2009 Lau: signaling is written “signalling” => please correct it.
-Table 1: 2007 Mas VR: the message is difficult to understand, maybe you can provide more information and delete a few numbers since it is not so relevant.
-Table 1: 2014 Talat RM: “increasing in PDGF plasma level in HCC” what does it mean? HCC patients have increased PDGF plasma levels? Please modify the message.
-lines 122-123: “Platelets sequester about 95% of total plasma serotonin in their dense granules, thus acting as a warehouse for serotonin, and release it in response to various stimuli [42]”. This sentence is partially copied from view of Aryal et al. 2018: Deciphering Platelet Kinetics in Diagnostic and Prognostic Evaluation of Hepatocellular Carcinoma. Volume 2018, Article ID 9142672 !
-line 131: TGFbeta1? Or just TGF?
-line 140: p-values are not required.
-line 144: space character missing
-line 145. Use??
-line 166: delete “blood”
-line 194: produced mediators??
-line 198: the? Correct it.
-line 198: Tumor cells have passed through the blood flow and have settled (homed) in a distant tissue. In the next section you start again with the statement that platelets protect tumor cells from shear forces in the blood flow and immune surveillance. This is difficult for the reader to follow, please modify the section.
-line 206: “platelet-initiated coagulation steps” platelets do not initiate coagulation steps, they rather contribute to coagulation induced by tumor cells.
-line207: Par1 and PAR2 and not “PAr”
-line 210: “All of these previously reported mechanisms are conveyed by selectins,” no, there are also further platelet adhesion receptors that contribute to a close interaction between platelets and tumor cells.
-line 214: also other adhesion receptors mediate aggregate formation
-line 217: “dependent tumor cell rolling along” tumor cells commonly do not role along the endothelium, because they do not express selectins.
-line 225: not all of the in vitro findings concerning platelet contribution to cancer or even HCC progression and metastasis have been confirmed in clinical studies. Please change the sentence.
-line 246: Candidates
-line 320: Please spend some words/explanations on the “European Setting”.
-line 361: “Eastern series” what does it mean?
-line 424: “All” not “Al”.
-line 425: “From” not “Form”
Author Response
Reviewer 1
The present review deals with the current knowledge of the impact of platelets on hepatocellular cancer progression and metastasis with special focus on PDGF, serotonin, EGF, and platelets as biomarkers in different clinical settings.
Response: We thank Reviewer#1 for the extraordinary work of correction and for the opportunity to improve the quality of the study.
Major issues:
1) The mechanisms by which platelets support cancer metastasis could be explained in more detail.
Response: According to the suggestions of Reviewer#1, we tried to improve the part dedicated to platelets and HCC metastasis.
2) figure 1: different pictures are integrate in the figure but the intention and content are hard to follow or to understand. In the introduction you refer to figure 1 but the different signaling pathways in the figure are not mentioned in the text. The figure should be modified completely and presented clearer, for instance, it is not clear where Growth Factors and Serotonin come from in the upper part of the figure.
Response: We agree with Reviewer#1. We divided the Figure in two different figures, trying to better explain the mechanisms reported. We also tried to report the different signaling pathways across the text
3) line 50: HCC not Hcc
Response: We thank Reviewer#1 for the extraordinary work of correction done. We systematically corrected all the refuses and errors across the text according to the Reviewer#1’s suggestions.
4) line 97: maybe one space character to much between PDGFR and levels.
Response: We modified accordingly.
5) line 100-103: you don´t need to indicate the p-values in a review article, that is rather uncommon.
Response: we removed the p-values as suggested.
6) line 104-107: the sentence is very long and difficult to understand. Please divide into two sentences.
Response: We divided the sentence accordingly.
7) line 112: delete “who”
Response: We modified accordingly.
8) Table 1: maybe you should not label it “learning poin”t, rather “result” or “outcome”.
Response: We modified accordingly.
9) Table 1: 2009 Lau: signaling is written “signalling” => please correct it.
Response: We modified accordingly.
10) Table 1: 2007 Mas VR: the message is difficult to understand, maybe you can provide more information and delete a few numbers since it is not so relevant.
Response: We modified the sentence.
11) Table 1: 2014 Talat RM: “increasing in PDGF plasma level in HCC” what does it mean? HCC patients have increased PDGF plasma levels? Please modify the message.
Response: We modified the sentence.
12) lines 122-123: “Platelets sequester about 95% of total plasma serotonin in their dense granules, thus acting as a warehouse for serotonin, and release it in response to various stimuli [42]”. This sentence is partially copied from view of Aryal et al. 2018: Deciphering Platelet Kinetics in Diagnostic and Prognostic Evaluation of Hepatocellular Carcinoma. Volume 2018, Article ID 9142672 !
Response: We rephrased the sentence as suggested.
13) line 131: TGFbeta1? Or just TGF?
Response: We modified accordingly.
14) line 140: p-values are not required.
Response: We modified accordingly.
15) line 144: space character missing
Response: We modified accordingly.
16) line 145. Use??
Response: We modified accordingly.
17) line 166: delete “blood”
Response: We modified accordingly.
18) line 194: produced mediators??
Response: We better clarified the sentence.
19) line 198: the? Correct it.
Response: We modified accordingly.
20) line 198: Tumor cells have passed through the blood flow and have settled (homed) in a distant tissue. In the next section you start again with the statement that platelets protect tumor cells from shear forces in the blood flow and immune surveillance. This is difficult for the reader to follow, please modify the section.
Response: We agree with Reviewer#1. We modified the section trying to make it easier for the reador to follow.
21) line 206: “platelet-initiated coagulation steps” platelets do not initiate coagulation steps, they rather contribute to coagulation induced by tumor cells.
Response: We modified accordingly.
22) line207: Par1 and PAR2 and not “PAr”
Response: We modified accordingly.
23) line 210: “All of these previously reported mechanisms are conveyed by selectins,” no, there are also further platelet adhesion receptors that contribute to a close interaction between platelets and tumor cells.
Response: We agree with Reviewer#1. We tried to implement this section.
24) line 214: also other adhesion receptors mediate aggregate formation
Response: We agree with Reviewer#1. We modified accordingly.
25) line 217: “dependent tumor cell rolling along” tumor cells commonly do not role along the endothelium, because they do not express selectins.
Response: We modified the sentence according to the suggestions of Reviewer#1, removing the part in which we reported the part in which we described the “dependent tumor cell rolling”.
26) line 225: not all of the in vitro findings concerning platelet contribution to cancer or even HCC progression and metastasis have been confirmed in clinical studies. Please change the sentence.
Response: We modified the sentence accordingly.
27) line 246: Candidates
Response: We modified accordingly.
28) line 320: Please spend some words/explanations on the “European Setting”.
Response: We entirely removed the phrase.
29) line 361: “Eastern series” what does it mean?
Response: We changed “Eastern series” in “Authors coming from Asian centers”.
30) line 424: “All” not “Al”.
Response: We modified accordingly.
31) line 425: “From” not “Form”
Response: We modified accordingly.
Reviewer 2 Report
This is a review paper on roles of platelets for HCC, which is well-written.Some minimum issues would be revised.
Page 3
Figure1. Letters in this figures are too small to read.
Page 5-8
Regarding EGFR and VEGFR, they are targets for systemic treatments for advanced HCC. If authors mention about platelets and systemic treatments using anti-EGFR pathways and VEGF/VEGFR, they are very informative and of value to understand platelets in HCC.
Page 7
It is an interesting issue that thrombocytosis is associated with worse prognosis in patients with HCC. If authors would show the survivals in patients with and without thrombocytosis, it is also informative.
Author Response
Reviewer 2
Comments and Suggestions for Authors
This is a review paper on roles of platelets for HCC, which is well-written. Some minimum issues would be revised.
Response: We thank Reviewer#2 for the very positive comments on the paper. We tried to improve the article according to the suggestions of Reviewer#2.
1) Page 3 Figure1. Letters in this figures are too small to read.
Response: We modified the figure, increasing the dimensions of the letters.
2) Page 5-8 Regarding EGFR and VEGFR, they are targets for systemic treatments for advanced HCC. If authors mention about platelets and systemic treatments using anti-EGFR pathways and VEGF/VEGFR, they are very informative and of value to understand platelets in HCC.
Response: The main intent of the present review was not to report in detail the mechanisms and the role of the different recently developed drugs for the treatment of HCC. Nevertheless, we agree with Reviewer#2 about the importance to stress the interconnection between platelets, tumor pathways, and the specific drugs developed for their blocking. Therefore, we reported at the end of the parts dedicated to EGFR and VEGFR the list of the anti-HCC drugs developed showing an anti-EGFR and/or anti-VEGFR mechanism.
3) Page 7 It is an interesting issue that thrombocytosis is associated with worse prognosis in patients with HCC. If authors would show the survivals in patients with and without thrombocytosis, it is also informative.
Response: We added Table 3 in which we reported some of the literature focused on thrombocytosis and worse results after the curative treatments resection and transplantation.
Reviewer 3 Report
Lai et al. reviewed the platelets is associated with HCC. Major problem of this manuscript is that the thrombocytopenia is often seen in patients with HCC. Authors should mention and discuss at this point.
Tejima K, Masuzaki R, Ikeda H, Yoshida H, Tateishi R, Sugioka Y, Kume Y, Okano T, Iwai T, Gotoh H, Katoh S, Suzuki A, Koike Y, Yatomi Y, Omata M, Koike K. Thrombocytopenia is more severe in patients with advanced chronic hepatitis C than B with the same grade of liver stiffness and splenomegaly. J Gastroenterol. 2010 Aug;45(8):876-84.
Author Response
Reviewer 3
Lai et al. reviewed the platelets is associated with HCC.
Response: We thank Reviewer#3 for the review and the comment.
1) Major problem of this manuscript is that the thrombocytopenia is often seen in patients with HCC. Authors should mention and discuss at this point.
Tejima K, Masuzaki R, Ikeda H, Yoshida H, Tateishi R, Sugioka Y, Kume Y, Okano T, Iwai T, Gotoh H, Katoh S, Suzuki A, Koike Y, Yatomi Y, Omata M, Koike K. Thrombocytopenia is more severe in patients with advanced chronic hepatitis C than B with the same grade of liver stiffness and splenomegaly. J Gastroenterol. 2010 Aug;45(8):876-84.
Response: We added the sentence commenting the impact of thrombocytopenia/portal hypertension as a negative effect on survival.
Round 2
Reviewer 1 Report
The authors have modified the manuscript thoroughly. The review can be accepted in the present form.
Reviewer 3 Report
All queries are fixed well.